# EgoChoir: Capturing 3D Human-Object Interaction Regions from Egocentric Views

**Yuhang Yang**[1], **Wei Zhai**[1†], **Chengfeng Wang**[1], **Chengjun Yu**[1], **Yang Cao**[1,2], **Zheng-Jun Zha**[1]

[1] University of Science and Technology of China

[2] Institute of Artificial Intelligence, Hefei Comprehensive National Science Center

https://yyvhang.github.io/EgoChoir

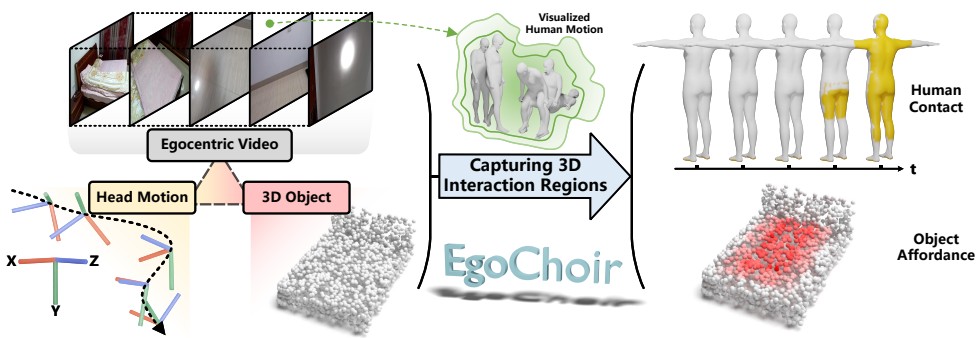

Figure 1: EgoChoir takes egocentric frames and head motion from head-mounted devices, along with the 3D object, to capture 3D interaction regions, including human contact and object affordance. The human motion is just visualized for intuitive observation of contact, yet it is not utilized by EgoChoir.

## Abstract

Understanding egocentric human-object interaction (HOI) is a fundamental aspect of human-centric perception, facilitating applications like AR/VR and embodied AI. For the egocentric HOI, in addition to perceiving semantics *e.g.*, "what" interaction is occurring, capturing "where" the interaction specifically manifests in 3D space is also crucial, which links the perception and operation. Existing methods primarily leverage observations of HOI to capture interaction regions from an exocentric view. However, incomplete observations of interacting parties in the egocentric view introduce ambiguity between visual observations and interaction contents, impairing their efficacy. From the egocentric view, humans integrate the visual cortex, cerebellum, and brain to internalize their intentions and interaction concepts of objects, allowing for the pre-formulation of interactions and making behaviors even when interaction regions are out of sight. In light of this, we propose harmonizing the visual appearance, head motion, and 3D object to excavate the object interaction concept and subject intention, jointly inferring 3D human contact and object affordance from egocentric videos. To achieve this, we present **EgoChoir**, which links object structures with interaction contexts inherent in appearance and head motion to reveal object affordance, further utilizing it to model human contact. Additionally, a gradient modulation is employed to adopt appropriate clues for capturing interaction regions across various egocentric scenarios. Moreover, 3D contact and affordance are annotated for egocentric videos collected from Ego-Exo4D and GIMO to support the task. Extensive experiments on them demonstrate the effectiveness and superiority of EgoChoir.

---

† Corresponding author.

38th Conference on Neural Information Processing Systems (NeurIPS 2024).

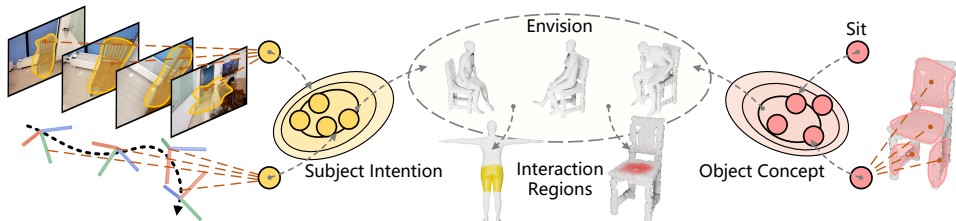

Figure 2: The subject intention, conveyed through synergistic visual appearances and head movements, along with the object interaction concept revealed by its structure and functionality, pre-formulate an interaction body image, which enables interaction regions to be envisioned.

# 1 Introduction

Human-object interaction (HOI) understanding aims to excavate co-occurrence relations and interaction attributes between humans and objects [91, 103]. For egocentric interactions, in addition to capturing interaction semantics like what the subject is doing or what the interacting object is [5, 25], knowing where the interaction specifically manifests in space *e.g.*, human contact [9, 80] and object affordance [16, 24] is also crucial. The precise delineation of spatial regions constitutes a pivotal component in numerous applications, like interacting with the scene in embodied AI [17, 71], interaction modeling in graphics [28, 95], robotics manipulation [52, 115], and AR/VR [11].

Most existing methods isolate the human and object to estimate contact or affordance regions [9, 29, 63, 80, 96, 101], capturing one aspect of interaction regions but neglecting the synergistic nature of interaction regions between interacting parties [59]. They delineate the region where objects should be operated without specifying the region of subjects intended for executing such operations, and vice versa. This oversight limits their efficacy in shaping final interactions. Some studies explore correlations between interacting parties to jointly estimate interaction regions for both the subject and object [34, 100, 102], in which observations of the interacting parties are quite crucial, whether appearances within exocentric visuals or compatible structures formed by geometries of the subject and object. However, the egocentric view possesses incomplete observations of interacting parties, for instance, when sitting on a chair or interacting with hands accompanied by head rotation, the interacting parties are only partially visible or even completely invisible. This leads to **ambiguity** between visual observations and interaction contents, which undermines the effectiveness of these methods, resulting in gaps when directly applied to egocentric scenarios.

Studies in cognitive science illustrate that humans make egocentric behaviors through coordination of the visual cortex, cerebellum, and brain to correlate visual observations, self-movement, and conceptual understanding, thereby revealing complementary interaction clues that link their embodiment and surroundings [1, 23, 66]. This motivates us to ponder: what clues could drive machines to capture effective interaction contexts and infer interaction regions from the egocentric view? Analogous to humans, in this paper, we propose harmonizing the visual appearance, head motion, and 3D object to infer 3D human contact and object affordance from egocentric videos (Fig. 1). Normally, objects are designed to fulfill certain human needs, the linkage between their functionalities and structures reveals their interaction concepts, implying the intention and interaction regions. When engaging in interactions with specific objects, visual observation synergistically changes with head movement, conveying the interaction intention [51]. The subject intention and object concept formulate an interaction "body image" [72, 75], with it, the region humans intend to contact, and the region that objects afford for such interactions could be pre-envisioned during forming the interaction (Fig. 2). This guides the estimation of interaction regions even when interacting parties move out of sight, eliminating the ambiguity between egocentric visual observations and interaction contents.

To consolidate the above insight, we present EgoChoir, a novel framework that integrates the visual appearance, head motion, and 3D object to excavate the object interaction concept and subject intention, collaboratively capturing 3D interaction regions. EgoChoir first links the semantic functionality and structures of the object by correlating interaction contexts within the appearance and motion with object geometry, thus mining the object interaction concept. Specifically, the appearance and motion features are mapped into interaction clues, and the object geometric feature, along with a semantic token that represents the functionality, queries these clues to calculate the 3D affordance through a parallel cross-attention. With affordance, the appearance feature is taken to

query complementary interaction clues from head motion and 3D affordance in parallel, excavating the subject intention and modeling the contact representation. Despite the framework being heuristic, egocentric interaction scenarios are quite distinct *e.g.*, with hand or body, which leads to varying effects of multiple interaction clues on modeling interaction regions in different scenarios. To adapt to this variability, EgoChoir employs modulation tokens to adjust gradients of specific layers that map interaction clues in the parallel cross-attention, endowing the model to adopt appropriate interaction clues for robustly estimating interaction regions across various egocentric scenarios.

Furthermore, we collect egocentric videos including 12 types of interactions with 18 different objects, and over $20K$ corresponding 3D object instances. 3D human contact and object affordance are also annotated for the collected data, which could serve as the first test bed for estimating 3D human-object interaction regions from egocentric videos. The key contributions are summarized as follows:

- We propose harmonizing the visual appearance, head motion, and 3D object to infer human contact and object affordance regions in 3D space from egocentric videos. It furnishes essential spatial representations for egocentric human-object interactions.
- We present EgoChoir, a framework that correlates complementary interaction clues to mine the object interaction concept and subject intention, thereby modeling the object affordance and human contact through parallel cross-attention with gradient modulation.
- We construct the dataset that contains paired egocentric interaction videos and 3D objects, as well as annotations of 3D human contact and object affordance. It serves as the first test bed for the task, extensive experiments on it demonstrate the effectiveness and superiority of EgoChoir.

## 2   Related Work

**Embodied Perception.** Embodied perception emphasizes actively understanding the surroundings and facilitates intelligent agents in learning and improving human-like skills through interactions [71]. This involves perceiving various attributes of the scene, *e.g.*, object functionality [18, 22, 58, 89, 98, 101, 109], scene semantics or geometry [15, 44, 36, 60, 61, 86], and sound [6, 7, 8, 21, 76]. Meanwhile, perceiving the embodied subject is also crucial, which involves anticipating the intention of the interacting subject [30, 83, 99] and the way to interact with objects [38, 82, 97, 115] or scene [28, 39, 50, 110]. These methods achieve significant progress in perceiving a certain side of the embodiment or surroundings. However, when embodied agents interact with their surroundings, the interaction manifests in both the interacting subject and the facing object. Capturing synergistic interaction between the interacting parties is crucial. EgoChoir aims to explore the synergy perception of interaction regions from egocentric videos, including human contact and object affordance.

**Egocentric Interaction Understanding.** So far, methods have made significant progress in several proxy tasks for understanding egocentric interactions, such as action recognition [33, 65, 77, 88], anticipation [53, 70, 93], moment query [41, 73], semantic affordance detection [47, 104, 106], and temporal localization [105, 107]. They endow machines to understand the semantic ("what") and temporal ("when") aspects of the interaction. Despite their importance in egocentric interaction understanding, the lack of spatial perception ("where") makes it challenging to form interactions in the physical world. Some methods explore grounding spatial interaction regions at the instance-level [2, 40, 114] or part-level [48, 49, 57], but only in 2D space, resulting in gaps when extrapolating to the real 3D environment. In contrast, EgoChoir captures the spatial aspect of egocentric interactions, and jointly estimates object affordance and human contact in 3D space.

**Perceiving Interaction Regions in 3D Space.** For 3D interaction regions, dense human contact [80] and 3D object affordance [16, 24] recently get much attention in the field. Methods estimate them typically follow two paradigms, one of which is to directly establish a mapping between geometries and semantics [28, 55, 56, 89, 96, 111], *e.g.*, "sit" links the seat of chairs, as well as the buttocks and thighs of humans. This paradigm establishes category-level connections between semantics and geometric regions, but it possesses limited generalization to unseen categories. Another paradigm explores correlations between geometries and interaction contents in 2D visuals *e.g.*, exocentric images [29, 62, 74, 80, 101, 102], taking correlations to guide the estimation. This endows the model to actively anticipate based on interaction contents, which generalizes better in unseen cases. However, incomplete observations in the egocentric view lead to ambiguous visual appearances for modeling the correlation, which affects their effectiveness. EgoChoir mitigates this influence by harmonizing multiple interaction clues that could provide effective interaction contexts.

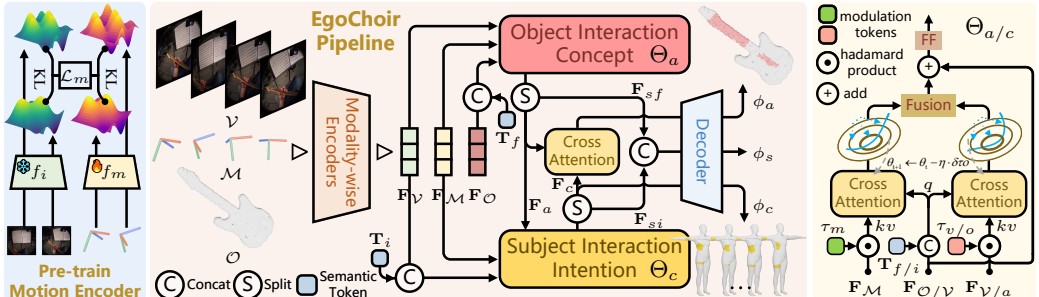

Figure 3: **Method.** EgoChoir first employs modality-wise encoders to extract features, in which the motion encoder is pre-trained by minimizing the distance between visual disparity and motion disparity. Then, it takes them to excavate the object interaction concept and subject intention, modeling the affordance and contact through parallel cross-attention with gradient modulation.

## 3 Method

The pipeline of EgoChoir is shown in Fig. 3, including extracting modality-wise features (Sec. 3.2), modeling the object affordance and human contact (Sec. 3.3), and the gradient modulation that enables to adopt appropriate clues to estimate interaction regions across various scenarios (Sec. 3.4).

### 3.1 Preliminaries

Given the inputs $\{\mathcal{V}, \mathcal{M}, \mathcal{O}\}$, where $\mathcal{V} \in \mathbb{R}^{T \times H \times W \times 3}$ indicates a video clip with $T$ frames of size $H \times W$, $\mathcal{M} \in \mathbb{R}^{T \times 12}$ denotes the translation vectors and rotation matrixes of head poses. $\mathcal{O} \in \mathbb{R}^{N \times 3}$ is an object point cloud with $N$ points. The goal is to learn a model $f$ that outputs temporal dense human contact $\phi_c \in \mathbb{R}^{T \times 6890 \times 1}$, 3D object affordance $\phi_a \in \mathbb{R}^{N \times 1}$, along with an interaction category $\phi_s$, expressed as: $\phi_c, \phi_a, \phi_s = f(\mathcal{V}, \mathcal{M}, \mathcal{O})$. 6890 is the number of SMPL [46] vertices.

### 3.2 Modality-wise feature extraction

Employing video backbones that are pre-trained by specific tasks like action recognition or contrastive learning [41] on egocentric datasets [13, 26] is a candidate approach to encode $\mathcal{V}$. However, we find that they tend to homogenize features across a sequence in our task (Sec. 4.3), this is detrimental to estimating temporally dynamic interaction regions. Thus, referring to HPS from videos [32, 35], EgoChoir adopts the paradigm that correlates per-frame features. Specifically, per-frame features are extracted through a pre-trained HRNet ($f_i$) [85], then, the joint space-time attention ($f_{st}$) is applied to establish temporal and spatial correlations among features, expressed as: $\mathbf{F}_{\mathcal{V}} = f_{st}(f_i(\mathcal{V})), \in \mathbb{R}^{TH_1W_1 \times C}$, where $H_1, W_1$ are height and width, $C$ is the feature dimension.

The relative change in head poses is a crucial clue for providing interaction contexts [37]. Thus, the relative head pose difference between each frame and the first frame is calculated, including translation difference $\bar{t}$ and rotation difference $\bar{R}$. It could be formulated as: $\bar{t}_i = t_i - t_0, \bar{R}_i = R_0^{-1} R_i$,

where $t_0, t_i \in \mathbb{R}^{1 \times 3}$ and $R_0, R_i \in \mathbb{R}^{3 \times 3}$ indicate the head translations and rotations at the first frame and $i$-th frame, $i \in [1, T]$. The calculated $\bar{t}$ and $\bar{R}$ are concatenated into the relative head motion $\bar{\mathcal{M}}$. Despite calculating relative changes in head pose, a motion encoder capable of encoding the variation is still needed. EgoChoir achieves this by associating the feature discrepancy between encoded motion features with the discrepancy in visual appearance features [79]. In detail, appearance features $\mathbf{F}_{\mathcal{V}}^j, \mathbf{F}_{\mathcal{V}}^k$ are extracted from two random frames in $\mathcal{V}$ by the frozen $f_i$, where $j, k$ means $j$-th, $k$-th frame and $j < k$. Then, the corresponding $j$-th, $k$-th head poses are selected from $\bar{\mathcal{M}}$ and encoded by $f_{\mathcal{M}}$ that is composed of MLP layers, obtaining motion features $\mathbf{F}_{\mathcal{M}}^j, \mathbf{F}_{\mathcal{M}}^k$. The $f_{\mathcal{M}}$ is trained by minimizing KL divergences calculated by $\mathbf{F}_{\mathcal{M}}^j, \mathbf{F}_{\mathcal{M}}^k$ and $\mathbf{F}_{\mathcal{V}}^j, \mathbf{F}_{\mathcal{V}}^k$, the loss can be formulated as:

$$\mathcal{L}_m = ||\sum_C \mathbf{F}_{\mathcal{M}}^i log(\epsilon + \frac{\mathbf{F}_{\mathcal{M}}^i}{\epsilon + \mathbf{F}_{\mathcal{M}}^j}) - \sum_{H_1W_1} \sum_C \mathbf{F}_{\mathcal{V}}^i log(\epsilon + \frac{\mathbf{F}_{\mathcal{V}}^i}{\epsilon + \mathbf{F}_{\mathcal{V}}^j})||_2, \quad (1)$$

where $\epsilon$ is a regularization constant. By constraining the distance between the visual discrepancies and motion discrepancies in feature space, the variation in appearance features moderately transmitter to

motion features, allowing $f_{\mathcal{M}}$ to extract motion features $\mathbf{F}_{\mathcal{M}} \in \mathbb{R}^{T \times C}$ with variations and associate with appearances. The object geometric feature $\mathbf{F}_{\mathcal{O}} \in \mathbb{R}^{N \times C}$ is extracted through the DGCNN [90]. Each encoder is further fine-tuned during the optimization of affordance and contact estimation.

### 3.3 Modeling object affordance and human contact

**Object interaction concept.** With the modality-wise features, EgoChoir correlates $\mathbf{F}_{\mathcal{V}}, \mathbf{F}_{\mathcal{M}}$ with $\mathbf{F}_{\mathcal{O}}$ to link the object functionality and structure, revealing the object interaction concept and calculating the affordance feature through parallel cross-attention. In specific, a semantic token $\mathbf{T}_f \in \mathbb{R}^{1 \times C}$ representing the functionality is concatenated with $\mathbf{F}_{\mathcal{O}}$ as the query, while $\mathbf{F}_{\mathcal{V}}, \mathbf{F}_{\mathcal{M}}$ are used as two parallel key-value pairs. In the parallel cross-attention, $\mathbf{F}_{\mathcal{V}}, \mathbf{F}_{\mathcal{M}}$ are scaled by learnable modulation tokens $\tau_v, \tau_m \in \mathbb{R}^C$. This modulates gradients of mapping layers and enables the model to extract effective interaction contexts from appropriate interaction clues across various scenarios, which is clarified in Sec. 3.4. The cross-attention is employed to model correlations among the query and key-value pairs parallelly, expressed as: $\bar{\mathbf{F}}_a = \Theta_a(\Gamma[\mathbf{T}_f, \mathbf{F}_{\mathcal{O}}], \tau_v \cdot \mathbf{F}_{\mathcal{V}}, \tau_m \cdot \mathbf{F}_{\mathcal{M}}), \in \mathbb{R}^{(N+1) \times C}$, where $\Theta_a$ denotes the transformer with parallel cross-attention, shown in Fig. 3, the fusion is composed of concatenation and MLP layers, $\Gamma$ indicates the concatenation, "·" is the hadamard product. $\bar{\mathbf{F}}_a$ is split into the affordance feature $\mathbf{F}_a \in \mathbb{R}^{N \times C}$ and semantic feature of the functionality $\mathbf{F}_{sf} \in \mathbb{R}^{1 \times C}$.

**Subject interaction intention.** As a manifestation of the object interaction concept, affordance implies the subject intention and assists in modeling intention semantics and human contact. With it, the $\mathbf{F}_{\mathcal{V}}$ queries complementary interaction clues from the motion feature $\mathbf{F}_{\mathcal{M}}$ and affordance feature $\mathbf{F}_a$ to derive the subject intention, and calculate the human contact and intention semantic features. Analogous to the affordance extraction, it can be expressed as: $\bar{\mathbf{F}}_c = \Theta_c(\Gamma[\mathbf{T}_i, \mathbf{F}_{\mathcal{V}} + pe_t], \tau_o \cdot \mathbf{F}_a, \tau_m \cdot (\mathbf{F}_{\mathcal{M}} + pe_t)) \in \mathbb{R}^{(TH_1W_1+1) \times C}$, where $\Theta_c$ is similar with $\Theta_a$, $\mathbf{T}_i \in \mathbb{R}^{1 \times C}$ is a token that represents the intention semantics, $pe_t \in \mathbb{R}^{T \times C}$ is a temporal position encoding which introduces temporal dynamics into human contact and it is expanded to $\mathbb{R}^{TH_1W_1 \times C}$. $\bar{\mathbf{F}}_c$ is split into semantic feature of the intention $\mathbf{F}_{si} \in \mathbb{R}^{1 \times C}$ and contact feature $\mathbf{F}_c \in \mathbb{R}^{TH_1W_1 \times C}$. Furthermore, to maintain the synergy between contact and affordance, $\mathbf{F}_c$ is then mapped to key-value features, and $\mathbf{F}_a$ queries the synergistic interaction regions from them through a cross-attention $f_{ca}$.

**Decoder.** The semantics of functionality and intention are correlated, thus, the $\mathbf{F}_{sf}, \mathbf{F}_{si}$ are concatenated to the semantic feature $\mathbf{F}_s$, then $\mathbf{F}_s$ is decoded into the categorical logits $\phi_s \in \mathbb{R}^n$ through MLP layers, $n$ is the number of interaction category. The affordance feature $\mathbf{F}_a$ is decoded in the feature dimension and projected to object affordance $\phi_a \in \mathbb{R}^{N \times 1}$. For the $\mathbf{F}_c$, in addition to decoding the feature dimension, the spatial dimension is mapped to the sequence of SMPL vertices. The human contact $\phi_c \in \mathbb{R}^{T \times 6890 \times 1}$ is output through two shallow MLP layers that decode the feature and spatial dimension. The overall loss is formulated as: $\mathcal{L} = \mathcal{L}_a + \mathcal{L}_c + \mathcal{L}_s$, where $\mathcal{L}_s$ is a cross-entropy loss, it constrains synergistic interaction semantics of the human and object. $\mathcal{L}_a$ and $\mathcal{L}_c$ optimize the affordance and contact respectively, both are a focal loss [42] plus a dice loss [54].

### 3.4 Gradient modulation

Egocentric interaction scenarios exhibit differences, *e.g.*, with hands or body, which affect the effectiveness of distinct interaction clue features for extracting interaction contexts in the parallel cross-attention. Assuming sitting down or operating with hands in the egocentric view, the former hardly observes interaction regions, in which the variation of head motion is a more effective clue for extracting interaction contexts. In contrast, the latter has less head movement but can derive rich contexts from the object interaction concept and visual appearances. Vanilla cross-attention presents limitations for adapting to diverse egocentric interactions.

Our goal is to enable the model to adopt appropriate interaction clues for modeling interaction regions across various egocentric scenarios. Some methods [20, 67, 87] balance information from distinct modalities by calculating the discrepancy of a consistent output (*e.g.*, category logits), they compute a scaling factor $\kappa$ from logits output by different modalities and take the $\kappa$ to modulate gradients in each modal branch, thereby adjusting the weight to balance each modality, it can be simplified as:

$$\theta_{t+1} \leftarrow \theta_t - \eta \cdot \kappa \cdot \frac{\partial \mathcal{L}}{\partial \theta_t}, \quad \kappa = \sigma(\frac{f_1(x_1)}{f_2(x_2)}), \tag{2}$$

where $\theta$ is the optimize parameter, $\eta$ is the learning rate, $\mathcal{L}$ is the loss function. $x_1, x_2$ represent inputs of distinct modalities, $f_1, f_2$ indicates layers and calculations to get the logits, and $\sigma$ denotes

Table 1: **Quantitative Results.** Metrics of baselines and ours on human contact and object affordance. The best results are covered with the mask, ◦ indicates the relative improvement to the first row.

| | Human Contact | | | | | Object Affordance | | |
|---|---|---|---|---|---|---|---|---|
| Method | Precision↑ | Recall↑ | F1↑ | geo. (cm)↓ | Method | AUC↑ | aIOU↑ | SIM↑ |
| BSTRO [29] | 0.42 | 0.45 | 0.42 | 43.21 | O2O [56] | 71.52 | 9.65 | 0.387 |
| DECO [80] | 0.54 ◦28% | 0.57 ◦26% | 0.53 ◦26% | 29.57 ◦31% | IAG [101] | 74.30 ◦3% | 11.21 ◦16% | 0.402 ◦4% |
| LEMON [102] | 0.65 ◦54% | 0.70 ◦55% | 0.67 ◦59% | 21.43 ◦50% | – | 75.97 ◦6% | 12.31 ◦27% | 0.410 ◦6% |
| Ours | **0.78** ◦85% | **0.79** ◦75% | **0.76** ◦81% | **12.62** ◦18% | – | **78.02** ◦9% | **14.94** ◦55% | **0.436** ◦13% |

an activation function. This manner seeks to equally weigh multi-modal inputs and avoid being dominated by a certain modality. While it differs from our expectations for the model, EgoChoir is expected to adopt appropriate clue features in different egocentric interaction scenarios by modulating gradients of specific layers. Actually, in addition to adding a scaling factor $\kappa$, the gradient can also be modulated by manipulating $\frac{\partial \mathcal{L}}{\partial \theta}$ in Eq. 2, which could be expanded into:

$$\frac{\partial \mathcal{L}}{\partial \theta_{mn}} = \frac{\partial \mathcal{L}}{\partial o_n} \frac{\partial o_n}{\partial z_n} \frac{\partial z_n}{\partial \theta_{mn}}, \quad \frac{\partial \mathcal{L}}{\partial o_n} \frac{\partial o_n}{\partial z_n} = \delta_n, \quad \frac{\partial z_n}{\partial \theta_{mn}} = o_m, \quad z_n = \theta_{mn} \cdot o_m + b, \quad (3)$$

where $o_m, z_n$ are input and output connected by a layer weight $\theta_{mn}$, $b$ is the bias, $o_n$ is the value of $z_n$ through an activation function. To clarify the formula, the partial derivative of $\mathcal{L}$ on certain $z$ is defined as $\delta$, taking the $\delta_n$ as an example, it represents the partial derivative of $\mathcal{L}$ with respect to $z_n$ in the subsequent layer that contains certain nodes. With the $\delta_n$, the first part of Eq. 3 can be rewritten as: $\frac{\partial \mathcal{L}}{\partial \theta_{mn}} = \delta_n \cdot o_m$, as seen from this formulation, scaling the feature $o_m$ also modulates the gradient, and this manner can primarily adjust the gradients of specific layers by selecting features to scale. Eventually, the parameters are updated as: $\theta_{t+1} \leftarrow \theta_t - \eta \cdot \delta \tau o$, where $\tau$ represents the learnable tokens to scale features in Sec. 3.3.

## 4 Experiment

### 4.1 Experimental setup

**Dataset. 1)** Source data: we collect video clips with egocentric interactions from Ego-Exo4D [27] and GIMO [113], encompassing over $300K$ frames across 12 interactions with 18 object categories. The paired ego-exo videos in Ego-Exo4D enable the annotation of human contact from exocentric views. GIMO includes aligned 3D human bodies and scene, the human contact can be calculated based on distance [29]. Besides, over $20K$ 3D object instances spanning 18 categories appearing in egocentric videos, are collected from multiple 3D datasets [14, 43, 81, 92, 94]. **2)** Annotation: we adopt a semi-automated approach to annotate human contact, involving multiple rounds of manual annotation and fine-tuning off-the-shelf model for inference. The calculated contacts in GIMO are also manually refined. Furthermore, we refer to the annotation pipeline [16, 102] to annotate the 3D object affordance. The statistical information about the dataset, including interaction categories distribution of video clips, the distribution of object affordance annotations, and the distribution of contact on different human body parts, are shown in Fig. 4.

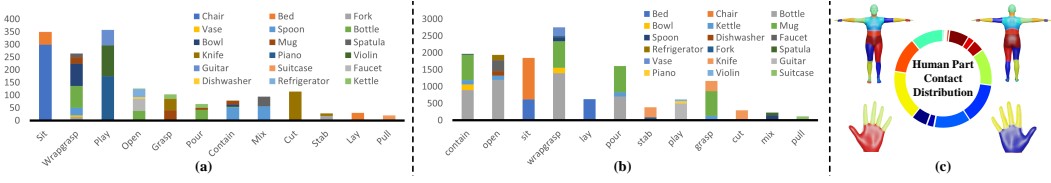

Figure 4: **Dataset Distribution. (a)** The distribution of different interaction categories and objects in video clips. **(b)** Category distribution of 3D object affordance annotation. **(c)** Distribution of contact annotations on human body parts.

**Annotation.** We annotate both 3D human contact and object affordance for the collected egocentric videos. Referring to DECO [80] and LEMON [102], the contact vertices are drawn on SMPL

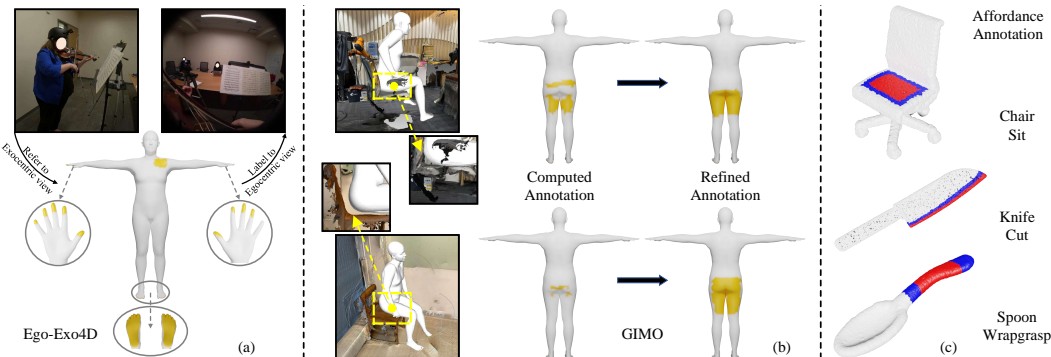

Figure 5: **Annotation of 3D human contact and object affordance.** **(a)** Annotate contact for data in Ego-Exo4D. **(b)** Contact annotation for GIMO dataset, including calculations and manual refinement. **(c)** 3D object affordance annotation, with the red region denoting that with higher interaction probability, while the blue region indicates the adjacent propagable region.

[46] through MeshLab [64], corresponding to the human contact in exocentric frames. For data in Ego-Exo4D, we select the best exocentric perspective and initially annotate the contact for 150 video clips, the process is shown in Fig. 5 (a). In detail, we select frames with a stride of 16 to manually annotate the contact and the remaining frames are consistent with the adjacent annotated frames. Next, annotators check per-frame annotations and refine those with slight changes. Then, We fine-tune LEMON [102] through the annotated contact, the human body needed by LEMON is obtained by SMPLer-X [4]. The remaining data is divided into groups for every 200 clips, and the fine-tuned model is used to predict human contact along with the manual refinement. Multiple rounds are conducted to obtain the final annotations. Please note that the annotations for each round are accumulated, and LEMON is fine-tuned each round, it takes exocentric frames as the input.

For data in GIMO, we first set a distance threshold [29, 31] to calculate the contact between the human body and the 3D scene, the threshold is set to $2cm$. However, we find that there is a deviation in the accuracy of human-scene alignment, which makes it hard to calculate all contacts using a unified threshold, shown in Fig. 5 (b). Besides, the scanned geometry cannot reflect deformation, which also affects the contact annotation. Therefore, we locate key frames of the interaction and visualize human bodies in the 3D scene for these frames, then manually refine the calculated contacts.

For 3D object affordance, shown in Fig. 5 (c), we annotate a high probability interaction region (red) and an adjacent propagable region (blue) on a 3D object, and calculate the 3D affordance annotation $S$ through a symmetric normalized laplacian matrix [16], formulated as:

$$S = (I - \alpha(D^{-0.5}WD^{-0.5})^{-1})Y, \quad W = 0.5(A + A^T), \quad A_{ij} = \begin{cases} \|\mathbf{v}_i - \mathbf{v}_j\|_2, & \mathbf{v}_j \in NN_k(\mathbf{v}_i) \\ 0, & \text{otherwise} \end{cases} \quad (4)$$

where $Y \in \{0, 1\}$ is the one-hot label vector and 1 indicates positive label, $\alpha$ is a hyper-parameter controlling decreasing speed, set to $0.995$. $A$ represents the adjacency matrix of sampled points in a KNN graph, $W$ is the symmetric matrix and $D$ is the degree matrix. $v$ is the $xyz$ spatial coordinate of the point in the red region and $NN_k$ denotes the set of $k$ nearest neighbors in the blue region.

**Metrics and baselines.** Referring to advanced work in estimating interaction regions [16, 29, 80, 101], the object affordance is evaluated through AUC, aIOU, and SIM. Precision, Recall, F1, and geodesic errors (geo.) are used to evaluate human contact estimation. Since there is no existing method to estimate 3D human contact and object affordance from the egocentric view, the constructed dataset is utilized to retrain methods that estimate interaction regions based on observations for comparison, including DECO [80], LEMON [102], etc. Note: some methods require certain modifications to their raw frameworks, details of metrics and each comparison method are provided in the appendix.

### 4.2 Experimental results

**Quantitative results.** Tab. 1 shows that our method outperforms baselines across all metrics in both human contact and object affordance estimation from egocentric videos. The gap between visual

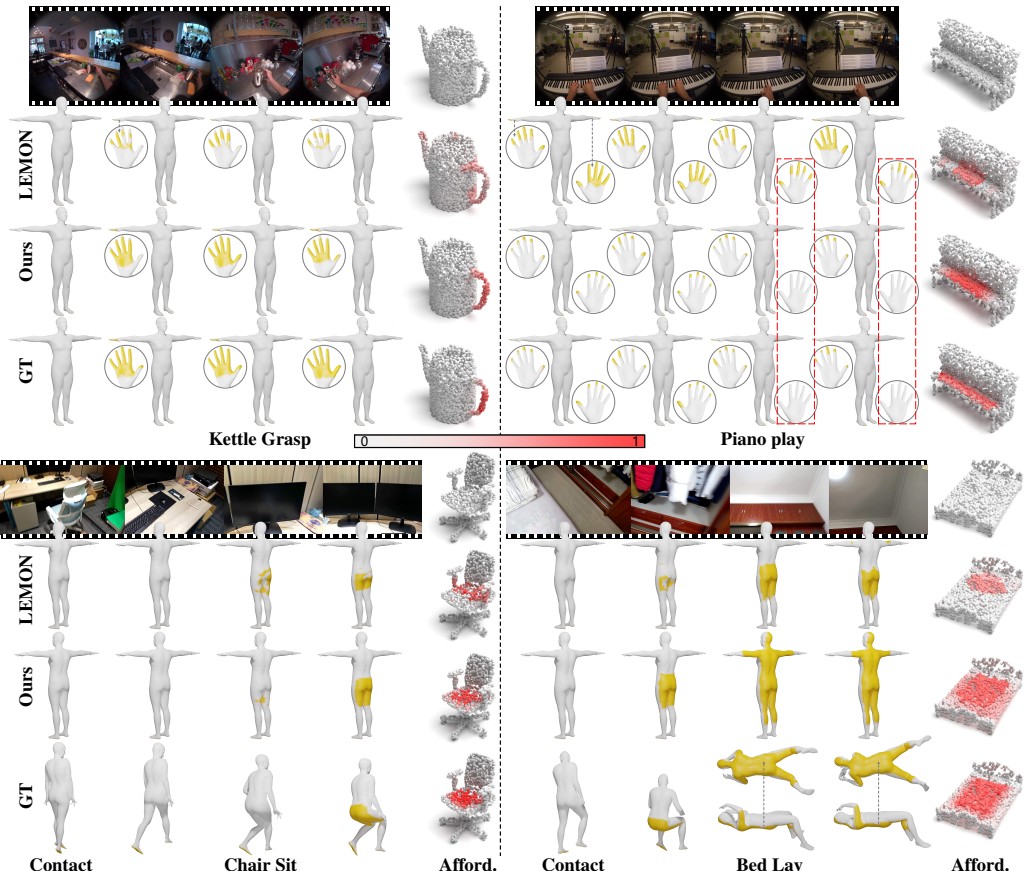

Figure 6: **Qualitative Results.** Contact vertices are colored yellow, and 3D object affordance are colored red, with the depth of red representing the affordance probability. Note: for intuitive visualization, the contact GT of body interactions are visualized on posed humans (last row) from GIMO [113]. Please zoom in for a better visualization and refer to the Sup. Mat. for video results.

appearance and interaction content, caused by incomplete observations of interacting parties, hinders the performance of methods that rely on visual cues, *e.g.*, BSTRO, DECO, O2O, and IAG, leading to suboptimal results for both contact and affordance. LEMON gets moderate results owing to modeling human-object geometric correlations. However, due to the parallel architecture of visual appearances and geometries in its framework, the incomplete appearance in the egocentric view diminishes its performance. In contrast, EgoChoir correlates interaction regions by linking the object interaction concept and subject intention, thereby bridging the gap and achieving better results. Semantics are primarily used to constrain the synergy of interaction regions and are not included in the main evaluation. The comparison of category prediction accuracy is reported in the appendix.

**Qualitative results.** Fig. 6 presents a qualitative comparison of contact and affordance estimated by our method and LEMON. As can be seen, our method yields more precise results and captures the temporal variation of contact, *e.g.*, playing the piano with two hands or one. Besides, in cases where the interaction regions are invisible (the below row), LEMON gets poor results due to the ambiguous guidance provided by visual observations. Our method adopts appropriate interaction clues to extract interaction contexts under different interaction scenarios and still infers plausible results.

## 4.3 Ablation study

We conduct a thorough ablation study to validate the effectiveness of the framework design and some implementation mechanisms, both quantitative and qualitative results are provided.

**Framework design.** The metrics when detaching certain framework designs are recorded in Tab. 2. The head motion $\bar{\mathcal{M}}$ and affordance $\mathbf{F}_a$ are crucial interaction clues to excavate the subject intention

Table 2: **Quantitative Ablations.** Metrics when detaching the head motion $\bar{\mathcal{M}}$, affordance $\mathbf{F}_a$ in $\Theta_c$, gradient modulation $\tau$, the $\mathbf{F}_s$, $f_{ca}$ for semantics and region synergy, and $pe_t$. As well as ablations of several implementations, including randomly initialize (ri.) $f_{\mathcal{M}}$ without pre-train, video extractors *e.g.*, SlowFast (S.F.) and Lavila (La.), divided space-time attention (d. $f_{st}$), ✗ means without.

| Metrics | Ours | ✗ $\bar{\mathcal{M}}$ | ✗ $\mathbf{F}_a$ | ✗ $\tau$ | ✗ $\mathbf{F}_s$ | ✗ $f_{ca}$ | ✗ $pe_t$ | ri. $f_{\mathcal{M}}$ | S.F. | La. | d. $f_{st}$ |
|---|---|---|---|---|---|---|---|---|---|---|---|
| **Prec.** | 0.78 | 0.68 | 0.71 | 0.72 | 0.74 | 0.72 | 0.75 | 0.73 | 0.67 | 0.70 | 0.76 |
| **Recall** | 0.79 | 0.73 | 0.64 | 0.71 | 0.71 | 0.75 | 0.78 | 0.69 | 0.77 | 0.79 | 0.75 |
| **F1** | 0.76 | 0.69 | 0.66 | 0.71 | 0.73 | 0.72 | 0.74 | 0.70 | 0.72 | 0.74 | 0.75 |
| **geo.** | 12.62 | 19.86 | 19.13 | 17.68 | 15.53 | 15.73 | 13.43 | 14.57 | 21.37 | 19.22 | 13.04 |
| **AUC** | 78.02 | 74.36 | 75.21 | 75.34 | 76.12 | 76.61 | 77.75 | 76.05 | 76.35 | 76.62 | 77.88 |
| **aIOU** | 14.94 | 11.75 | 12.05 | 12.36 | 13.04 | 13.63 | 12.86 | 12.92 | 12.52 | 13.10 | 14.62 |
| **SIM** | 0.436 | 0.403 | 0.410 | 0.413 | 0.422 | 0.425 | 0.429 | 0.423 | 0.422 | 0.427 | 0.431 |

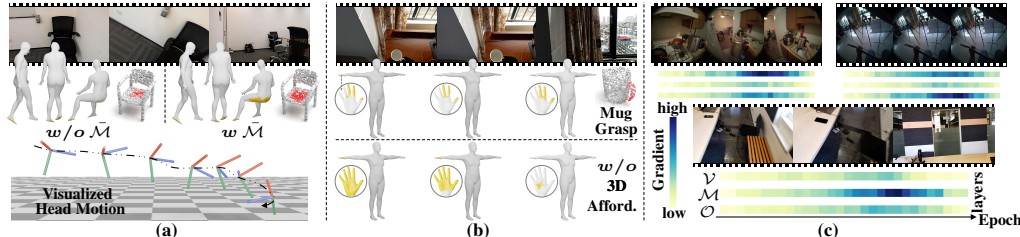

Figure 7: **Qualitative Ablations.** **(a)** Results of the human contact and object affordance $w/o$ and $w$ head motion, along with the visualized head motion. **(b)** The lack of 3D affordance leads to over-prediction and temporal inconsistency of human contact. **(c)** Gradients of layers mapping different interaction clues under distinct input interactions during sampled 30 training epochs.

and object interaction concept, the performance significantly declines without them. The gradient modulation $\tau$ enables the model to robustly adapt to various interaction scenarios, the absence of this mechanism also impacts performance. The semantic feature $\mathbf{F}_s$ and the $f_{ca}$ establish semantic and regional synergy between interacting parties, the metrics drop when detaching any of them. The temporal position encoding $pe_t$ introduces disparity in temporal dimension and eliminates some false positives, removing it decreases the precision and aIOU.

Additionally, qualitative results are provided for further analysis. Fig. 7 (a) demonstrates the results $w$ and $w/o$ head motion, as observed, the model can hardly anticipate interaction regions without the head motion, particularly for body interactions. The ablation of $\mathbf{F}_a$ in modeling human contact is shown in Fig. 7 (b), which shows that the 3D affordance constrains the contact scope and maintains the temporal coherence of contact. Even if the object disappears in some frames, the model still plausibly infers based on the interaction concept provided by 3D affordance. The effectiveness of gradient modulation is illustrated in Fig. 7 (c). As can be seen, the gradients of layers mapping different interaction clues in the parallel cross-attention exhibit significant differences across inputs with distinct interactions, indirectly reflecting that the modulation endows the model to adopt appropriate clues for interaction context modeling and generalize to various interaction scenarios.

**Implementation mechanisms.** The metrics of some implementation mechanisms are also shown in Tab. 2. Randomly initializing the motion encoder makes it difficult to capture variations in motion features, adversely affecting the extraction of interaction contexts and resulting in a performance decline. For the extraction of video features $\mathbf{F}_{\mathcal{V}}$, we also test video backbones such as SlowFast [19], Lavila [112] pre-trained on egocentric datasets [13, 26]. The precision and recall of contact estimation reveal that they tend to predict consistent results across all frames (see qualitative results in appendix), leading to lower precision. Divided space-time attention [3] is also implemented to replace the joint one, while the joint space-time attention demonstrates superior performance.

## 4.4 Performance analysis

Here, we outline several heuristic attributes of the model that can robustly reason interaction regions from egocentric videos, and provide insights for further improving the model performance.

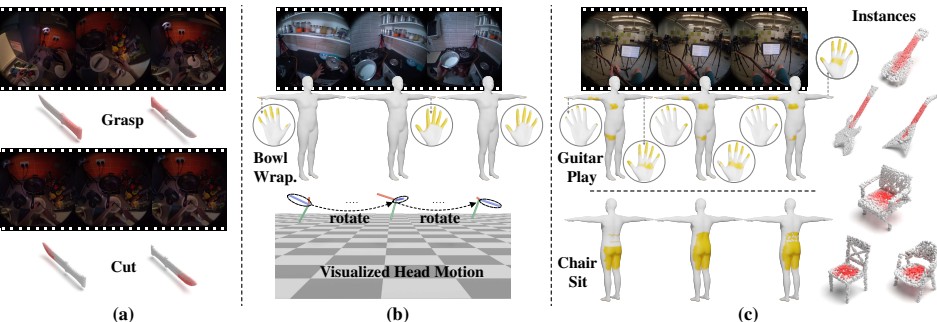

Figure 8: **Analysis. (a)** The changing interaction contents correspond to dynamic 3D object affordances *e.g.*, from grasp to cut. **(b)** Dynamic contact, *e.g.*, left-right change can be implied by the head rotation. **(c)** Results of distinct object categories and different instances in one interaction scenario. Note that slight differences exist in contact results with different object instances, but overall consistency, one of the inferred contacts is visualized. Wrap. means wrapgrasp.

**Dynamic region.** Human contact and object affordance regions vary as the interaction evolves. We conduct an experiment to validate whether the model captures this attribute. Fig. 8 (a) shows the results of changing object affordances and Fig. 8 (b) demonstrates the estimated changing human contact, unlike methods that distinguish left-right hands by masks or boxes, EgoChoir leverages the head motion *e.g.*, rotations, for inference. This provides a way to get rid of intermediary models.

**Multiplicity.** The multiplicity is another crucial attribute of the interaction, encompassing interacting with multiple objects and instances. This requires the model to differentiate interactions with distinct objects and generalize across various instances. As shown in Fig. 8 (c), our method infers credible interaction regions with different objects and instances, which indicates that our model effectively captures interaction contexts with specific objects. It completes the estimation through the interaction contexts rather than merely mapping to specific categories or instances.

**Whole-body motion.** Recently, significant progress has been made in estimating human pose from the egocentric view [12, 37, 84], facilitating the capture of egocentric whole-body

Table 3: Metrics when using whole-body motion.

| Precision | Recall | F1 | geo. | AUC | aIOU | SIM |
|---|---|---|---|---|---|---|
| 0.80 | 0.82 | 0.79 | 11.24 | 78.54 | 15.46 | 0.448 |

motion. We test replacing motion features in the existing framework with global geometric features derived from a sequence of human bodies (SMPL vertices), and find that the performance continues to improve, shown in Tab. 3. This validates the effectiveness of harmonizing multiple clues for estimating interaction regions and suggests the potential for boosting performance in the future.

## 5   Discussion and conclusion

We propose harmonizing the visual appearance, head motion, and 3D object to infer 3D human contact and object affordance regions from egocentric videos. It furnishes spatial representation of the interaction to facilitate applications like embodied AI and interaction modeling. Through the constructed data and annotations, we train EgoChoir, a novel framework that mines the object interaction concept and subject intention, to estimate object affordance and human contact by correlating multiple interaction clues. With the gradient modulation in parallel cross-attention, it adopts appropriate clues to extract interaction contexts and achieves robust estimation of interaction regions across diverse egocentric scenarios. Extensive experiments show that EgoChoir could infer dynamic and multiple egocentric interactions, as well as its superiority over existing methods. EgoChoir offers fresh insights into the field and facilitates egocentric 3D HOI understanding.

**Limitations and future work.** Currently, EgoChoir may estimate the interaction region slightly before or after the exact contact frame, possibly due to a lack of spatial relation perception between interacting parties. Future work could consider incorporating 3D scene conditions and estimated whole-body motion [37, 108] to better constrain the spatial relation, achieving more fine-grained estimation of interaction regions and promoting egocentric human-scene interaction modeling.

**Acknowledgments**. This work is supported by the National Natural Science Foundation of China (NSFC) under Grants 62306295 and 62225207.

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

# Appendix

## A Implementation Details

### A.1 Method details

**Visual encoder.** The backbone to extract visual features is HRNet-W40 pre-trained on the ImageNet, which takes $\mathcal{V}$ as the input and outputs the feature with the shape of $\mathbb{R}^{T \times 2048 \times 7 \times 7}$, then, a $1 \times 1$ convolution kernel is employed to obtain $\mathbf{F}'_{\mathcal{V}} \in \mathbb{R}^{T \times 768 \times 7 \times 7}$. In our implementation, T is set to 32, $C$ in the main paper is 768, and $H_1, W_1$ are both 7. $\mathbf{F}'_{\mathcal{V}}$ is flattened to $\mathbb{R}^{T \cdot 49 \times 768}$ to enter a transformer with joint space-time attention $f_{st}$. The depth of $f_{st}$ is set to 2, it contains 12 heads with each head dimension 64, the mapping dimension in the FeedForward is 768. Through the $f_{st}$, we obtain the feature $\mathbf{F}_{\mathcal{V}} \in \mathbb{R}^{T \cdot 49 \times 768}$.

**Motion encoder.** The motion encoder $f_{\mathcal{M}}$ is composed of MLP layers, projecting a sequence of head poses to the feature space, and obtaining the motion feature $\mathbf{F}_{\mathcal{M}} \in \mathbb{R}^{T \times 768}$. For the training of the motion encoder $f_{\mathcal{M}}$, there is also an online training mechanism, which directly adds $\mathcal{L}_m$ to the whole loss function $\mathcal{L}$ for end-to-end training. However, this mechanism carries the risk of model collapse, when the visual backbone is being fine-tuned, the KL divergence among visual features varies, this causes a situation where $\mathcal{L}_m$ optimizes the KL divergence between visual features to 0, then, the KL divergence between motion features also tends to 0, affecting the entire model. We also conduct an experiment to validate this point. Please refer to Sec. C.3.

**Point encoder.** The DGCNN is employed to extract the geometric features of 3D objects. The number of nearest neighbor points in KNN is 20. We fuse local graph features with the global feature to obtain the geometric feature $\mathbf{F}_{\mathcal{O}} \in \mathbb{R}^{N \times 768}$, in our implementation, $N$ is set to 2000.

**Transformer with parallel cross-attention.** We take $\Theta_a$ as an example to introduce the implementation details of the transformer with parallel cross-attention, and $\Theta_c$ is similar to $\Theta_a$. $\tau_v, \tau_m, \tau_o \in \mathbb{R}^{768}$ are learnable tokens to scale the appearance, motion, and geometry features respectively. In $\Theta_a$, the $\mathbf{F}_{\mathcal{V}}, \mathbf{F}_{\mathcal{M}}$ multiply with $\tau_v, \tau_m$ before mapping to $kv$ vectors. $\mathbf{F}_{\mathcal{O}}$ is mapped to the query and $\mathbf{F}_{\mathcal{V}}, \mathbf{F}_{\mathcal{M}}$ after scaling are mapped to two key-value pairs, the query vector performs cross-attention with these two key-value pairs separately. The cross-attention also contains 12 heads, each head with a dimension of 64, and the dropout ratio is 0.1. The fusion layer is composed of MLP layers to fuse query features, obtaining the affordance feature $\mathbf{F}_a \in \mathbb{R}^{N \times 768}$. The way to calculate $\mathbf{F}_c \in \mathbb{R}^{T \cdot 49 \times 768}$ is similar, but there are some details that need to be clarified. In $\Theta_c$, the $\mathbf{F}_{\mathcal{O}}, \mathbf{F}_{\mathcal{M}}$ are multiplied with $\tau_o, \tau_m$ and mapped to the key-value pairs, $\mathbf{F}_{\mathcal{V}}$ is mapped to the query. Besides, $\mathbf{F}_{\mathcal{V}}$ and $\mathbf{F}_{\mathcal{M}}$ are added with a temporal position encoding $pe_t \in \mathbb{R}^{T \times 768}$, $pe_t$ only applies to the temporal dimension, and the spatial dimension at each time step of $\mathbf{F}_{\mathcal{V}}$ adds the same value.

**Decoder.** The affordance feature $\mathbf{F}_a$ and semantic feature $\mathbf{F}_s$ concatenated by $\mathbf{F}_{sf}, \mathbf{F}_{si}$ are decoded directly through MLP layers, obtaining the object affordance $\phi_a \in \mathbb{R}^{2000 \times 1}$ and semantic interaction category $\phi_s \in \mathbb{R}^{12}$. For the contact feature $\mathbf{F}_c$, the decoding of spatial and feature dimensions is decoupled. Specifically, the feature dimension is first mapped from 768 to 1, representing the probability of contact, and then the spatial dimension is mapped to the vertex sequence of SMPL, ultimately obtaining the human contact $\phi_c \in \mathbb{R}^{T \times 6890 \times 1}$. DECO [80] decodes the contact by directly mapping the feature dimension to 6890. We also conduct the ablation experiment and find that the decoupled decoding works better, please refer to Sec. C.3.

**Loss.** Here, we further clarify the loss function. For the output $\phi_a, \phi_c, \phi_s$, each one has the corresponding ground truth $\hat{\phi}_a, \hat{\phi}_c, \hat{\phi}_s$. The $\mathcal{L}_s$ is a cross-entropy calculated through $\phi_s$ and $\hat{\phi}_s$, the $\mathcal{L}_a, \mathcal{L}_c$ possess the same formulation, expressed as:

$$
\begin{aligned}
\mathcal{L}_{a/c} =& 1 - \frac{\sum_j^N yx + \epsilon}{\sum_j^N y + x + \epsilon} - \frac{\sum_j^N (1-y)(1-x) + \epsilon}{\sum_j^N 2 - y - x + \epsilon} \\
&+ \frac{1}{N} \sum_j^N [-(1-\alpha)(1-y)x^\gamma \log(1-x) - \alpha y(1-x)^\gamma \log(x)],
\end{aligned}
\tag{5}
$$

where $N$ indicates the number of points or vertices in $\phi_a$ or $\phi_c$, $x$ is the prediction $(\phi_a, \phi_c)$, $y$ is the ground truth $(\hat{\phi}_a, \hat{\phi}_c)$, $\epsilon$ is set to $1e$-10, $\alpha, \gamma$ are set to 0.25 and 2 respectively.

Table 4: The collected 12 different interactions with 18 different objects. Obj. indicates objects, Int. denotes interactions, wrap. is wrapgrasp and Refrige. is Refrigerator.

| Obj. | Bottle | Kettle | Bowl | Bed | Fork | Faucet | Guitar | Chair | Dishwasher |
|---|---|---|---|---|---|---|---|---|---|
| **Int.** | wrap. open contain pour | grasp open contain pour | wrap. contain | sit lay | wrap. stab | open | play | sit | open |

| Obj. | Mug | Knife | Spoon | Spatula | Piano | Violin | Vase | Suitcase | Refrige. |
|---|---|---|---|---|---|---|---|---|---|
| **Int.** | wrap. grasp pour contain | grasp cut stab | wrap. mix contain | wrap. mix | play | play | wrap. | pull | open |

## A.2 Training and inference

EgoChoir is implemented by PyTorch and trained with the Adam optimizer. The training epoch is set to $100$, the training batch size is set to $8$, and the initial learning rate is $1e$-4 with cosine annealing. All training processes are on 2 NVIDIA A40 GPUs (20 GPU hours). The HRNet backbone is initialized with the weights pre-trained on ImageNet.

For each video, 32 frames are sampled for training. To maximize data usage and maintain a degree of randomness, the start frame is randomly selected from the first $n$-32 frames, where $n$ is the total frame number of the video, and the last frame is the $n$-th frame of the video, 32 frames are uniformly sampled between the start and end frame. The corresponding head poses are also indexed for training. This ensures that the sampled frames basically cover the whole process of an interaction. Additionally, for each video sample, a 3D instance corresponding to the category of interacting object in the video is randomly selected for training. This strategy helps the model generalize across instances while allowing for many-to-many combinations between videos and 3D objects, enhancing the diversity of training samples. Note that for videos involving multiple interacting objects, the ground truth of contact is selected based on the input 3D object category, which enables the model to estimate object-specific interaction regions. During the inference, the entire video is segmented into clips, each containing 32 frames. The last clip will be padded with the final frame if it has fewer than 32 frames. Each clip is paired with the same 3D object, allowing the inference of interaction regions for the whole video, while preserving the dynamic nature of both contact and affordance.

# B Dataset Construction

## B.1 Collection

We collect videos that contain egocentric hand and body interactions from Ego-Exo4D [27] and GIMO [113], encompassing 12 different interactions with 18 different objects, shown in Tab. 4. The original video has a long duration, which either contains too much redundant interaction content or content without interaction context for a long time. Therefore, we segment the collected videos to ensure that each clip has clear interaction contents, with a duration of 5-15 seconds. Both Ego-Exo4D and GIMO provide head trajectories. Trajectories from GIMO are aligned with the video frames, while trajectories in Ego-Exo4D are sampled at $1k$ HZ. Therefore, we select the middle one of every 33 head poses as the head pose which aligns with the video frames. Besides, we collect over $20k$ 3D object instances from multiple 3D object datasets [14, 43, 81, 92, 94], corresponding to categories of interacting objects in collected egocentric videos. Ultimately, we construct a dataset containing 1570 egocentric video clips, exceeding $300K$ frames, which can be trained in a many-to-many combination manner with over $20K$ 3D instances. Among them, 1216 video clips are used for training, and 354 are used for testing. Note: to validate the model's generalization ability to unseen scenes, the egocentric scene in the training set and test set are almost non-overlapping.

# C  Experiments

Here, we provide a further detailed explanation of the experimental setup, including the baselines and evaluation metrics. Besides, additional experimental results are also provided.

## C.1  Baselines

**BSTRO** [29]: BSTRO concatenates downsampled vertices of the SMPL template onto the image features extracted by HRNet, then it employs a multi-layer transformer to estimate the contact vertex. We take egocentric frames to train BSTRO with its mask mechanism, the vertex of SMPL is downsampled to $431$.

**DECO** [80]: DECO utilizes two branches with cross-attention to parse the human part and scene semantics in images, thus facilitating the estimation of human contact. Following its instructions, we take the Mask2Former [10] to create scene segmentation maps for egocentric frames and use the scene and contact branches to train the DECO, the HRNet is used as the image backbone, align with the encoder used in the EgoChoir.

**O2O-Afford** [80] O2O-Afford aims to ground the 3D affordance through the object-object interaction relation. It calculates the correlation between different 3D object features accompanied by a spatial distance constraint. Although the input format differs from our setting, the insight of this method can still be used to form a comparison. The key modulation is we take pixel features of egocentric images as the kernel to slide over sampled seed point features of the object, obtaining content-aware seed point features. The pixel features, content-aware seed point features and the object global feature are aggregated and upsampled as the final representation of object affordance.

**IAG-Net** [101]: IAG-Net anticipates object affordance by establishing the correlations between interaction contents in the image and the geometric features of object point cloud. We use Grounded-SAM [69] to get the bounding boxes of interacting subject and object, and train IAG-Net by inputting egocentric frames.

**LEMON** [102]: LEMON correlates human and object geometries with images to jointly estimate 3D human contact, object affordance and their relative spatial relation, it needs posed human bodies as the input. We directly use the provided humans for cases in the GIMO dataset and take SMPLer-X [4] to estimate posed humans from exocentric frames in Ego-Exo4D. The posed human body, egocentric images, and 3D objects are used as inputs to train LEMON. The curvature of humans and objects needed by LEMON is calculated by Trimesh and the software Cloudcompare. Note that the layers used to predict relative human-object spatial relation in the original LEMON structure are removed.

## C.2  Evaluation metrics

The metrics for evaluating human contact prediction include Precision, Recall, F1, and geodesic distance. Precision is the ratio of correctly predicted positive observations to the total predicted positives and measures the accuracy of the positive predictions made by a model. Recall is the ratio of correctly predicted positive observations to all observations in the actual class and measures the ability of a model to capture all the positive instances. F1-score is the harmonic mean of Precision and Recall. It provides a balance between Precision and Recall, making it a suitable metric when there is an imbalance between classes. They could be formulated as:

$$Precision = \frac{TP}{TP + FP}, Recall = \frac{TP}{TP + FN}, F1 = \frac{2 \cdot Precision \cdot Recall}{Precision + Recall}, \tag{6}$$

where $TP$, $FP$, and $FN$ denote the true positive, false positive, and false negative counts, respectively. The geodesic distance is utilized to translate the count-based scores to errors in metric space. For each vertex predicted in contact, its shortest geodesic distance to a ground-truth vertex in contact is calculated. If it is a true positive, this distance is zero. If not, this distance indicates the amount of prediction error along the body.

Object affordance are evaluated by AUC [45], aIOU [68] and SIM [78]. The Area under the ROC curve, referred to as AUC, is the most widely used metric for evaluating saliency maps. The saliency map is treated as a binary classifier of fixations at various threshold values (level sets), and an ROC curve is swept out by measuring the true and false positive rates under each binary classifier. IoU is the most commonly used metric for comparing the similarity between two arbitrary shapes. The IoU

Table 5: Metrics of LEMON and our method for each interaction category. Prec. indicates Precision, wrap. is wrapgrasp.

| | Metrics | grasp | open | lay | sit | wrap. | pour | pull | play | stab | contain | cut | mix |
|---|---|---|---|---|---|---|---|---|---|---|---|---|---|
| **LEMON** | Prec. | 0.68 | 0.80 | 0.34 | 0.45 | 0.72 | 0.79 | 0.80 | 0.64 | 0.73 | 0.72 | 0.77 | 0.78 |
| | Recall | 0.72 | 0.77 | 0.42 | 0.53 | 0.69 | 0.72 | 0.28 | 0.58 | 0.64 | 0.75 | 0.80 | 0.68 |
| | F1 | 0.68 | 0.78 | 0.36 | 0.48 | 0.70 | 0.74 | 0.41 | 0.61 | 0.66 | 0.72 | 0.79 | 0.71 |
| | geo. | 29.52 | 18.15 | 41.62 | 37.31 | 25.73 | 13.47 | 17.23 | 14.95 | 19.82 | 23.75 | 11.90 | 21.39 |
| | AUC | 58.01 | 72.12 | 55.16 | 58.92 | 61.79 | 39.77 | 54.05 | 66.34 | 47.37 | 49.84 | 74.38 | 61.57 |
| | aIOU | 2.30 | 6.15 | 2.23 | 3.19 | 5.91 | 3.25 | 1.22 | 6.36 | 1.93 | 3.17 | 4.04 | 6.40 |
| | SIM | 0.17 | 0.18 | 0.09 | 0.15 | 0.42 | 0.15 | 0.04 | 0.27 | 0.20 | 0.25 | 0.31 | 0.29 |
| **Ours** | Prec. | 0.75 | 0.88 | 0.6 | 0.80 | 0.80 | 0.86 | 0.87 | 0.72 | 0.80 | 0.80 | 0.87 | 0.85 |
| | Recall | 0.80 | 0.83 | 0.74 | 0.80 | 0.74 | 0.73 | 0.33 | 0.71 | 0.88 | 0.75 | 0.89 | 0.73 |
| | F1 | 0.75 | 0.82 | 0.62 | 0.79 | 0.74 | 0.77 | 0.48 | 0.70 | 0.83 | 0.76 | 0.87 | 0.77 |
| | geo. | 25.28 | 12.74 | 26.60 | 7.52 | 21.4 | 6.7 | 6.94 | 4.22 | 13.68 | 17.80 | 3.49 | 14.14 |
| | AUC | 62.06 | 75.02 | 81.07 | 90.89 | 64.79 | 46.51 | 57.08 | 75.03 | 51.22 | 52.68 | 83.64 | 66.18 |
| | aIOU | 5.33 | 8.83 | 19.48 | 36.89 | 8.91 | 6.75 | 7.32 | 9.56 | 3.92 | 6.02 | 8.02 | 8.34 |
| | SIM | 0.24 | 0.23 | 0.56 | 0.62 | 0.56 | 0.43 | 0.21 | 0.30 | 0.26 | 0.38 | 0.50 | 0.40 |

measure gives the similarity between the predicted region and the ground-truth region, and is defined as the size of the intersection divided by the union of the two regions. It can be formulated as:

$$IoU = \frac{TP}{TP + FP + FN}. \tag{7}$$

The similarity metric (SIM) measures the similarity between the prediction map and the ground truth map. Given a prediction map $P$ and a continuous ground truth map $Q^D$, $SIM(\cdot)$ is computed as the sum of the minimum values at each element, after normalizing the input maps:

$$(P, Q^D) = \sum_i min(P_i, Q_i^D), \quad where \sum_i P_i = \sum_i Q_i^D = 1. \tag{8}$$

### C.3 Additional results

Here, we provide more quantitative and qualitative experimental results to further demonstrate the effectiveness and superiority of the method.

**Metrics for each category.** The overall metrics are provided in the main paper, metrics for each category of LEMON and our method are reported in Tab. 5. It can be seen that our method outperforms the comparison baseline across almost all categories. For body interactions like "sit" and "lay", the results are much better than the baseline. This is because these interaction scenarios have significant ambiguity between visual observation and the interaction content. Observation-based methods struggle to predict plausible results, while our method overcomes this ambiguity by extracting effective interaction context from appropriate interaction clues, leading to better results.

**Detach the foot contact.** For human contact, in most scenarios, the feet are in contact with the ground, while certain hand contact regions are relatively small. This results in a situation where the model, even if it only predicts foot contact, could achieve favorable evaluation metrics. To further validate the model's performance of contact estimation, we retrain the model and comparison baselines without considering foot contact, reported in Tab 6. As can be seen, our method still exhibits the best performance.

Table 6: Contact metrics when detaching the foot contact.

| | Precision | Recall | F1 | geo. |
|---|---|---|---|---|
| BSTRO | 0.33 | 0.29 | 0.29 | 54.24 |
| DECO | 0.48 | 0.45 | 0.44 | 37.62 |
| LEMON | 0.52 | 0.50 | 0.50 | 31.27 |
| Ours | 0.67 | 0.63 | 0.65 | 22.67 |

**Additional quantitative results.** As illustrated in Sec. A.1, we conduct an experiment to validate the performance when training the motion encoder $f_{\mathcal{M}}$ online, directly adding $\mathcal{L}_m$ to the entire loss $\mathcal{L}$, the metrics are reported in Tab. 7.

For the head motion, we also attempt to calculate the relative head pose between adjacent two frames as $\bar{\mathcal{M}}'$, in this manner, the frame sampling strategy during training requires a slight modification. Because only by sampling consecutive frames does the head pose make sense. Specifically, the

Table 7: Metrics when training $f_{\mathcal{M}}$ online (adding $\mathcal{L}_m$ to $\mathcal{L}$), using adjacent relative head pose $\bar{\mathcal{M}}'$, and directly decode ($\triangleright$) $\mathbf{F}_c$ at feature dimension. As well as the error bar (e.b.) of EgoChoir.

| | Precision | Recall | F1 | geo. (cm) | AUC | aIOU | SIM |
|---|---|---|---|---|---|---|---|
| $+\mathcal{L}_m$ | 0.73 | 0.75 | 0.72 | 16.76 | 76.32 | 12.05 | 0.423 |
| $\bar{\mathcal{M}}'$ | 0.74 | 0.71 | 0.71 | 15.59 | 76.23 | 12.72 | 0.425 |
| $\triangleright\,\mathbf{F}_c$ | 0.75 | 0.76 | 0.74 | 15.10 | – | – | – |
| e.b. | $0.78 \pm 0.02$ | $0.79 \pm 0.01$ | $0.76 \pm 0.01$ | $12.62 \pm 1.5$ | $78.02 \pm 0.5$ | $14.94 \pm 0.3$ | $0.436 \pm 0.02$ |

Figure 9: More results inferred by our method. For egocentric body interactions, the whole-body motion is visualized for intuitive observation.

strategy for obtaining the start frame remains unchanged, and the subsequent $T$-1 frames are directly selected for training after obtaining the start frame. The results are reported in Tab. 7.

The result of directly decoding the feature dimension of $\mathbf{F}_c$ to 6890 is also shown in the table, the performance is inferior to decoupling spatial and feature dimensions.

The metrics in the main paper and appendix report the best results, we also provide the error bar of EgoChoir as a reference, shown in Tab. 7. For the prediction of interaction categories, we compare the $top$-1 accuracy with video backbones like SlowFast [19] and Lavila [112]. $Acc\_1$ for SlowFast is 0.79, Lavila is 0.84, and ours is 0.80, our method still performs comparably well.

**Additional qualitative results.** More qualitative results are visualized in Fig. 9, involving various hand and body interactions with distinct objects from the egocentric videos. As mentioned in the main paper, using pre-trained video backbones tends to homogenize the features across a sequence, resulting in almost the same contact estimation for the whole sequence. We visualize the human contact when taking Lavila [112] as the backbone to extract $\mathbf{F}_{\mathcal{V}}$, as shown in Fig. 10, in this case, contact is predicted even before it actually occurs.

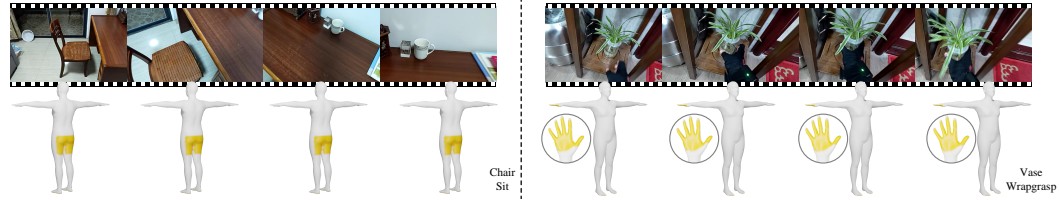

Figure 10: The estimated human contact when taking Lavila [112] as the backbone to extract video features. In this case, the contact is predicted even before it actually occurs.

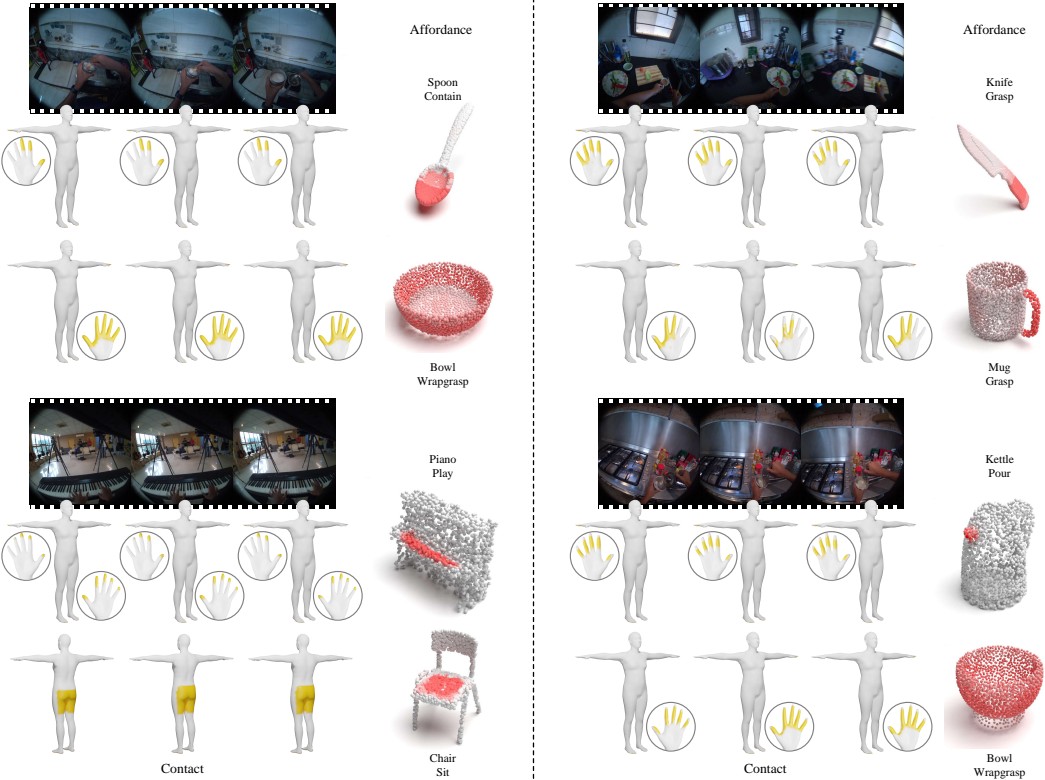

Figure 11: More results that demonstrate interactions with multiple object categories, including the human contact and object affordance.

Furthermore, we present more results showcasing interactions with multiple object categories, as depicted in Fig. 11. The ability to infer the affordance and contact associated with specific objects when interacting with multiple objects is crucial, which facilitates interaction modeling with various objects, and assists embodied agents in operating specific objects.

## D    Society Impact

The method unleashes the ability to estimate 3D interaction regions from the egocentric view, benefiting downstream applications such as embodied AI and interaction modeling. However, it currently cannot cover all interaction types, which may lead to confusing predictions in certain interactions and introduce risks to the entire system.

