# OpenReview forum: "EgoChoir: Capturing 3D Human-Object Interaction Regions from Egocentric Views"
_NeurIPS.cc/2024/Conference — NeurIPS 2024 poster_

### Official Review · Reviewer_L6qS · 2024-06-28

**Soundness:** 3
**Presentation:** 2
**Contribution:** 2
**Rating:** 6
**Confidence:** 3

**Summary:**

This paper describes an approach to estimate interaction between humans and objects from egocentric videos. To this end, head movement, 3D object meshes and 2D egocentric video are used as input and processed individually before being combined to predict contact regions in both object and subject. Experiments are carried out on Ego-Exo4D and GINO datasets, with additional annotations provided for both.

**Strengths:**

+ novel model in which the flow of information makes sense
+ quantitative and qualitative results give a significant amount of insight into the method's performance and inner workings. It's clear that all modules contribute to the increased performance over baselines.

**Weaknesses:**

- contributions of the paper a bit unclear, in terms of novelty over related work
- critical discussion missing
- paper hard to read

Contribution: The approach seems to be heavily influenced by LEMON, and the comparisons in the experiment section are also predominantly with LEMON. Still, differences between the two models are not outlined. It is only when really reading through the text and the appendix that differences in architecture and (training) details because more clear. But it's left to the reader. Linking Figures 1-3 more tightly with the text can help to better understand the particular novelties of the method. Now, these figures are relatively disjoint from the story.

Critical discussion missing: I would have appreciated a more critical discussion. While statements on Lavila's anticipation bias are reported, I don't see critical reflections on the performance reported for the main model. Still, given that human-object contact is modeled, the results depicted in Fig. 8 (for example) raise a number of questions, such as why the blade of the knife is identified (rather than the handle), or why the 3D model of the bottle is off, and why one of the two hands is not considered in the interaction with the bottle). Also, from the ablation study, the influence of various key components seems marginal. Leaving out many seemingly important terms still leaves a performance well higher than LEMON. Is there a very strong prior encoded? The experimental section gives me the feeling that the authors have been more critical to other works than to their own. This raises questions about the merits.

Paper hard to read: I found the paper particularly verbose, and it seems the authors have prioritized colorful language over concise, precise descriptions. The result of that choice is that many parts, including abstract and introduction, are particularly hard to read. Given the complexity of the work and the sheer number of concepts and symbols, it's easy to get confused when structure and language are not to-the-point. As a result, it's difficult to understand the true contributions of the work. Evidently, there is a host of related work on capturing estimating contact in HOI, also from the first person perspective. What seems to be novel is the more connected processing of the inputs, including head motion. But I found little motivation into the specific design choices to do so. For one, it's not motivated why a geometry-semantics approach is not preferred. While it's clear that this only works for objects that have been considered during training, it seems the used data features objects that are rather standard, with affordances that can be well predicted.

**Questions:**

1. What is the (technical) novelty of the current approach over related works (in particular LEMON)?
2. What is the influence of a learned prior on the contact regions of both object and subject? Could the authors show the variation in estimations across videos of the same interaction?

**Limitations:**

Partly. While some limitations have been addressed, it's clear that the system is rather specific in the way it's been evaluated.

---

> ### Author Rebuttal · Authors · 2024-08-05
>
> > **W1. Contributions, in terms of novelty over related work.**
>
> We summarize the types of existing methods that estimate interaction regions. A distinct type involves estimating 2D regions [1]. For methods in 3D, the types include: 1) part-level contact [2]; 2) estimate human or object in isolation [3]; 3) estimate regions for a single exocentric image [4]; 4) estimate from the egocentric view, but only focus on hands [5].
>
> Different from them, we propose estimating **3D** interaction regions at dense **vertex-level** for **both** humans and objects from **egocentric videos**. Additionally, we extend egocentric estimation to the **body**. In this case, interacting subjects and objects may disappear from view, the model should address ambiguity between visual observations and interaction contents, which is more challenging.
>
> [1] Detecting human-object contact in images
> [2] CHORE: Contact, Human and Object REconstruction from a single RGB image
> [3] DECO: Dense estimation of 3D human-scene contact in the wild
> [4] LEMON: Learning 3D Human-Object Interaction Relation from 2D images
> [5] ARCTIC: A Dataset for Dexterous Bimanual Hand-Object Manipulation
>
> ---
> > **W2. Differences with LEMON.**
>
> * **Task and interaction clues**: LEMON estimates interaction regions for an exocentric image, where humans are observed relatively intact, and human mesh recovery (HMR) methods can be employed to obtain human geometries. It leverages clues such as the object and posed human geometries, as well as the interaction content (semantics) in images. EgoChoir estimates the regions for egocentric videos (temporally dense human contact). From this view, appearances alone may not provide effective interaction contexts, and human geometry is also hard to obtain. Thus, EgoChoir combines head movement, visual appearance, and 3D object as clues to model interaction regions. Note: a significant performance gap remains between HMR methods on egocentric and exocentric views, addressing the concern of "why a geometry-semantics approach is not preferred."
> * **Technical differences**: LEMON regards clues (semantics and geometries) as equally important for modeling. Its core technique involves building semantic correlations between geometries using image content as a shared key-value and constructing geometric correlations through a handcraft feature: curvature. In contrast, Egochoir seeks to differentiate the effectiveness of multiple clues (e.g. appearance, motion) based on interactions (hand or body). It achieves this by introducing a gradient modulation mechanism in the parallel cross-attention that correlates these clue features.
>
> ---
> Based on responses to **W1/W2**, we briefly reiterate contributions:
>
> * We first propose estimating interaction regions of the whole human body and objects from egocentric videos.
> * Introducing a heuristic framework EgoChoir. Analogous to human behavior, it harmonizes the visual appearance, head motion, and 3D object to model interaction regions and adapt to distinct egocentric interaction scenarios through a gradient modulation mechanism.
> * Annotating contact for collected video clips and affordance for 3D objects, which supports the training and evaluation of the task.
>
> ---
> > **W3. The experiments mentioned and critical discussion.**
>
> 1) **Comparisons are predominantly with LEMON**: We report quantitative comparison metrics with five baselines in Tab. 1. LEMON performs better than other baselines, so we just chose it for qualitative comparison.
> 2) **Critical discussion missing**: In lines 317-321, we critically clarify that the current model may estimate the region slightly before or after the exact frame due to the lack of spatial relation perception. This problem is currently difficult to solve because recovering human meshes from an egocentric view is challenging. So that the spatial relation is hard to capture.
> 3) **Marginal influence of components, leaving out still better than LEMON**: In Tab.2, some metrics, e.g. precision and F1, are shown in percentages, the improvement is actually significant. For instance, head motion $\bar{\mathcal{M}}$ increases the precision from 0.68 to 0.78, a 10-percentage point. Plus, without certain key designs, e.g. object concept $F_{a}$, some metrics (F1:0.66, AUC:75.21) are not higher than LEMON (F1:0.67, AUC:75.97). LEMON relies on image content as guidance, while in egocentric videos, the subject and object disappear in many frames. Its framework is hard to address in this case and leads to lower performance. That's why EgoChoir leaves out some designs still better than it.
> 4) **Questions about Fig. 8**: The definition of object affordance is key to understanding the case in Fig. 8. Affordance is first defined as "opportunities of interaction" [1]. In the computer vision field, object affordance refers to the functional regions that support certain interactions [2], it contains implicit functional semantics and is not completely equivalent to the contact region. In Fig. 8, videos display cutting with a knife and pouring from a bottle, so affordance regions are the blade and top of the bottle.
> 5) **Only works for standard objects**: The 3D objects we collect also include those from scanning and multi-view reconstruction. We present inference results on these noisy 3D objects **in the added PDF**.
> 6) **Learned prior? Same interaction across videos**: EgoChoir actually learns to reasonably adopt appropriate clue features based on distinct egocentric interactions, to capture effective interaction contexts. These clues are complementary, enabling it to handle situations such as appearance variations and missing subjects or objects. We provide qualitative results about the same interaction across videos **in the added PDF**, and quantitative results are reported in Appendix Tab. 5.
>
> [1] The ecological approach to visual perception
> [2] 3D AffordanceNet: A Benchmark for Visual Object Affordance Understanding

---

> > ### Comment · Reviewer_L6qS · 2024-08-13
> > **Post-rebuttal feedback**
> >
> > Thank you for addressing my questions (and those of the other reviewers). I now understand better the contributions of the paper. I would really appreciate if those, and the other ambiguous issues I mentioned, can be clearly stated in the paper. That would make it much easier to read, and will facilitate the appreciation of the work. I have increased my rating.

---

> > > ### Author Response · Authors · 2024-08-14
> > >
> > > Thanks for providing the feedback and increasing the rating. We will integrate your suggestions into the final version.

---

> ### Author Response · Authors · 2024-08-13
>
> Dear reviewer:
> As we approach the final day of the discussion phase, we hope to know whether our response has addressed your concerns to merit an increase in the rating, or if there are any issues that you would like us to clarify. Thanks for your time and consideration.

---

### Official Review · Reviewer_fLMx · 2024-07-10

**Soundness:** 3
**Presentation:** 3
**Contribution:** 3
**Rating:** 6
**Confidence:** 4

**Summary:**

This paper deals with the problem of inferring human-object interaction regions from egocentric views. It tackles the challenge of incomplete observations of interacting parties in the egocentric view by integrating the information from the visual appearance, head motion, and 3D object. It jointly infers 3D human contact and object affordance by exploring parallel cross-attention between the two parts. Moreover, 3D
contact and affordance are annotated for egocentric videos collected from existing datasets to evaluate the task.

**Strengths:**

1) The paper first tackles the task of estimating 3D contact and affordance from egocentric view by harmonizing multiple interaction clues including visual appearance, head motion, and 3D object.
2) A new framework is proposed to jointly infer human contact and object affordance regions in 3D space from egocentric videos. In the framework, object affordance and human contact are modeled through parallel cross-attention with gradient modulation to deal with the challenge of incomplete visual appearance in egocentric view.
3)  To evaluate the new task, a dataset is constructed that contains paired egocentric interaction videos and 3D objects, as well as annotations of 3D human contact and object affordance.  Extensive experiments are conducted on the constructed dataset to demonstrate the effectiveness of the proposed framework.

**Weaknesses:**

Several important information regarding the technical details and the dataset are missing.  For example,
1) Since a new dataset is constructed to evaluate the new task, the statistical information about the dataset (e.g., the proportations of interaction categories, ratio of contact and affordance annotations for different body parts and object categories) should be presented.
2) In Section 3.4, gradient modulation is achieved with learnable tokens to scale features to adapt to different scenarios. However, it is not clear whether these learnable tokens depend on input features. If yes, how? If not, then the tokens are fixed once the training is over and it is not possible to adapt to new scenarios during testing.

**Questions:**

1) The latter part of equation (4) seems unnecessary and even confusion. Actually, it is easy to understand that scaling the features would affect the gradient on the model parameters.
2) The right part of Figure 3 is difficult to understand. More details are needed to explain the notations and procedure.
3) Writing needs further refinement. For example,
    a) line-158, Obejct -> Object
    b) line-186, ...that decoupled "decode" the feature...
    3) line-233, punctuations error

**Limitations:**

Yes.

---

> ### Author Rebuttal · Authors · 2024-08-05
>
> > **W1. The statistical information about the dataset.**
>
> We follow the advice and provide statistical information about the dataset **in the newly added PDF**, including interaction categories distribution of video clips, the distribution of object affordance annotations, and the distribution of contact on different human body parts. We will add this information in the future version.
>
> ---
>
> > **W2. Modulation tokens that scale features to adapt to different scenarios.**
>
> The model is expected to adaptively capture effective interaction contexts from different interaction clues when inputting distinct scenarios (body or hand interaction). Different input scenarios result in distinct distributions of clue features, such as the appearance $F_\mathcal{V}$ and head motion $F_\mathcal{M}$. To achieve adaptation, the model should differentiate which clue feature is effective for the input scenario based on their feature distributions. However, such specific feature distribution is hard to capture. Therefore, we employ learnable tokens to enhance or weaken certain feature dimensions (e.g. for hand interaction, some dimensions in $F_\mathcal{V}$ represent hand features, while for body interaction, certain dimensions in $F_\mathcal{M}$ express significant head translation and rotation), we provide a schematic figure about this **in the newly added PDF**. By doing so, clues exhibit significant differences in feature distribution across distinct interaction scenarios. It simultaneously adjusts gradients of mapping layers in the parallel cross-attention, making the layers handle the clue feature discrepancies of different input scenarios. During testing, the key to adapting to different scenarios is the learned mapping layers, the key-value input of the parallel cross-attention depends on the combination of input clue features and modulation tokens, these tokens densely scale clue features (at feature dimension) that are extracted from different scenarios, aligning the feature distribution that the mapping layers handled. Thus, these layers differentiate the utilization of multiple clue features (e.g. whether querying from appearance, object concept, or head motion) under distinct scenarios, achieving the adaption.
>
> ---
>
> > **Q1-3. The latter part of Eq. 4 and the right part of Figure 3.**
>
> * The latter part of Eq.4 may indeed cause confusion, we will remove it and integrate the former part of Eq. 4 into Eq. 3 to make it more concise and clearer. Additionally, we will correct the typo you mentioned, thanks again for pointing it out.
>
> * The right of Fig. 3 actually represents two modules $\Theta_{a}$ and $\Theta_{c}$, which model affordance and contact respectively. They have the same model structure but different parameters and inputs. $\Theta_{a}$ corresponds to object interaction concept and $\Theta_{c}$ corresponds to subject interaction intention (middle of Fig. 3). As described in Sec. 3.3, $\Theta_{a}$ takes 3D object feature $F_\mathcal{O}$ as the query, head motion $F_\mathcal{M}$ and appearance $F_\mathcal{V}$ as two parallel key-value pairs to calculate the affordance feature. For $\Theta_{c}$, the $F_\mathcal{V}$ is the query, while affordance feature $F_{a}$ and $F_\mathcal{M}$ are parallel key-value pairs, it calculates the contact feature. Both $\Theta_{a}$ and $\Theta_{c}$ perform the gradient modulation. We will add notations in Fig.3 and more details in Sec. 3.3 to clarify the procedure.

---

> > ### Comment · Reviewer_fLMx · 2024-08-14
> >
> > Thanks for the clarifications which have addressed my concerns. After reading other reviewers' comments, I would like to keep my original rating.

---

### Official Review · Reviewer_paDU · 2024-07-14

**Soundness:** 3
**Presentation:** 3
**Contribution:** 3
**Rating:** 7
**Confidence:** 4

**Summary:**

This paper investigates inferring 3D human contact and object affordance from a combination of egocentric video, human head motion, and 3D object point cloud. Inspired by real human behavior, which is based on visual observations, self-movement, and conceptual understanding, the authors propose a framework called EgoChoir. This framework utilizes modality-wise encoders to extract features from different input modalities and then infers object interaction concepts and subject intentions. The authors also construct a dataset comprising paired egocentric interaction videos (from EgoExo4D and GIMO) and 3D objects as the first test bed, demonstrating that EgoChoir outperforms existing baselines in inferring 3D Human-Object Interaction Regions.

**Strengths:**

- The paper is well-written, with a clear and concise background and motivation.
- The key idea is ingenious and intuitive, drawing inspiration from real human behaviors and making it easy to understand.
- The empirical evaluation is comprehensive, covering a range of existing baselines and two recent datasets EgoExo4D and GIMO, demonstrating the effectiveness of the proposed approach.

**Weaknesses:**

- The authors could leverage the existing annotations in EgoExo4D, such as 3D hand pose and scene annotation, to improve the quality of synchronization between the annotated affordance and video. Since most human-object interactions in EgoExo4D involve only two hands, utilizing these annotations could enhance the accuracy of human contact and object affordance annotation.
- The authors mention that the egocentric scenes in the training and test sets are "almost non-overlapping," but the meaning of this term is unclear. Could the authors clarify whether "non-overlapping" refers to: no exact same videos, no same scenes or no same activities (e.g., training on cooking videos and evaluating on music videos). And how did they determine this train-test split.
- It would be beneficial to provide more elaboration on the application examples of the proposed method, e.g. illustrating its potential use cases.

**Questions:**

Please refer to the weaknesses part.

**Limitations:**

The authors discuss the limitations in Section 5.

---

> ### Author Rebuttal · Authors · 2024-08-05
>
> > **W1. Leverage the existing annotations in EgoExo4D, such as 3D hand pose and scene annotation.**
>
> Thanks for the advice. Combining 3D hand and body poses with the scene could indeed improve the annotation accuracy, which enables the contact to be calculated through the spatial distance at first, with only manual refinement needed. However, while building the dataset, we found that not all data in EgoExo-4D have these pose annotations, which hinders us from making a unified annotation workflow. As a result, we ultimately adopt a semi-automated annotation approach. In the future, as the dataset gradually updates the annotations, we will consider scaling up the dataset in this way.
>
> Plus, integrating 3D hand or body poses with the scene can potentially improve the prediction accuracy of the region in temporal. As we mentioned in the limitations (Sec. 5), due to the lack of spatial relation between the subject and object, the current model may estimate the interaction region slightly before or after the exact frame, while 3D poses and scenes can provide this spatial relation, offering a potential solution.
>
> ---
>
> > **W2. Explanation of the scenes "non-overlapping."**
>
> Here, the "scenes" refer to the background. Both the EgoExo-4D and GIMO datasets include the same interaction in different backgrounds (e.g. cut food in different kitchens). We realize that randomly splitting the training and test sets causes interaction clips with the same background to appear in both sets. In this case, the evaluation metrics cannot reflect whether the model is reasoning based on background information or interaction contents. Therefore, we manually split the training and test sets to ensure that clips with the same interaction have different backgrounds in two sets. This partition tests whether the model is inferring based on interactions and its generalization across different backgrounds.
>
> ---
>
> > **W3. Elaboration on the application examples.**
>
> Thanks for the constructive suggestion. We list some potential application scenarios:
>
> 1) Interaction Modeling: some studies [1,2] provide datasets to facilitate interaction reconstruction and generation from egocentric videos. These tasks usually encounter problems such as misalignment, floating, and penetration of the interacting subject and object in spatial. Our method provides a spatial surface constraint, such as the subject contact and object affordance. These representations can be extracted from our method as a condition for those modeling methods to improve the authenticity and spatial rationality of the interaction.
> 2) Embodied AI: learning skills from human demonstrations is an important part of embodied intelligence [3]. A feasible solution to achieve this is to first parse elements related to interactions from the embodied perspective (egocentric) demonstration, including the contact body part and object functional region. Then, retargeting the perceived elements to the agent's own configuration (e.g. dexterous hand) and facing objects. Taking the middle-level spatial perception to drive low-level control signals to complete the interaction.
>
> We will add a section about applications in the future version.
>
> [1] OAKINK2 : A Dataset of Bimanual Hands-Object Manipulation in Complex Task Completion
> [2] HOI4D: A 4D Egocentric Dataset for Category-Level Human-Object Interaction
> [3] DexMV: Imitation Learning for Dexterous Manipulation from Human Videos

---

> > ### Comment · Reviewer_paDU · 2024-08-14
> >
> > Thanks for the clarifications. After reading all the reponses and other reviewers' comments, I am inclined to keep my original score.

---

### Official Review · Reviewer_PyYF · 2024-07-15

**Soundness:** 4
**Presentation:** 4
**Contribution:** 3
**Rating:** 6
**Confidence:** 4

**Summary:**

This egocentric paper, aims to capture 3D interactions such as human contact and objects affordance, to achieve this it uses head motion, 3d object and the visual appearance

**Strengths:**

The add to Ego-Exo-4D the interaction data through a semi-automated process
The approach appears to beat other works on this newly labelled data.
There is extensive analysis of the results around the backbones

**Weaknesses:**

How accurate is the semi-automated annotation process
It's not clear where the right-hand side of Figure 3 is in the central system diagram
The fusion of the modality-wise encoders is basically a few cross-attention layers
It would be interesting to quantify the importance of each modality to the process

**Questions:**

The work provides good performance and is clearly written, what is the key contribution to the model fusion network?

**Limitations:**

nothing more than whats written above

---

> ### Author Rebuttal · Authors · 2024-08-05
>
> > **W1. How accurate is the semi-automated annotation process?**
>
> As described in Appendix B.2, during the semi-automated annotation process, we conduct a manual check and refinement after each round of model prediction to ensure the accuracy of contact annotations. This is because the model estimates approximate regions in the initial round but performs limitedly in fine-grained regions. To further validate the accuracy of contact annotations (semi-automated part: Ego-Exo4D), we conduct a cross-dataset validation. Specifically, we train LEMON to estimate contact in two settings using __exocentric__ images:
> 1) train on 3DIR dataset [1] and test on annotated Ego-Exo4D part
> 2) train on Ego-Exo4D part and test on 3DIR
>
> We use overlap object categories in the two datasets for testing, and the evaluation metrics of the two groups are shown in the table:
> |      | Precision | Recall |  F1  | geo. (cm) |
> |:----:|:---------:|:------:|:----:|:---------:|
> | 1    | 0.72      | 0.73   | 0.73 |   12.32   |
> | 2    | 0.70      | 0.74   | 0.71 |   13.14   |
>
> Note: the contact annotations in 3DIR are done entirely manually and are relatively more accurate. Meanwhile, the two groups are close in terms of metrics, confirming the accuracy of semi-automated annotations. This also suggests a potential way to scale up the dataset.
> [1] LEMON: Learning 3D Human-Object Interaction Relation from 2D images
>
> ---
>
> > **W2. It's not clear where the right-hand side of Figure 3 is in the central system diagram.**
>
> Sorry for the confusion. Actually, the right of Fig. 3 represents two modules that have the same model structure but different parameters and inputs. We denote them as $\Theta_{a}$ and $\Theta_{c}$. $\Theta_{a}$ is in the "Object Interaction Concept," while $\Theta_{c}$ is in the "Subject Interaction Intention", modeling affordance and contact, respectively. We will add notations of these modules to the central part of Fig. 3.
>
> ---
>
> > **W3/Q1. The key contribution to the model fusion network. Quantify the importance of each modality to the process.**
>
> The core to fuse modality-wise features lies in what information flow is used to combine them and adapt appropriate clue features (e.g. appearance, head motion, or object concept), to model spatial regions for distinct egocentric interaction scenarios (e.g. body or hand). In our fusion model, the object feature $F_\mathcal{O}$ is taken to capture interaction context from the appearance $F_\mathcal{V}$ and head motion $F_\mathcal{M}$, modeling the affordance region. The appearance feature $F_\mathcal{V}$ then queries complementary interaction contexts from the mined object concept (affordance $F_{a}$) and $F_\mathcal{M}$ to model contact region. Parallel cross-attention is a technical mechanism that completes the above process. In addition, unlike employing vanilla cross-attention directly, we introduce gradient modulation in the parallel cross-attention, enabling the model to adapt to various egocentric interaction scenarios.
>
> Regarding the importance of each modality, Tab. 2 in the main paper shows the quantitative impact of different modalities, we copy the modality-related results to the table below. It includes conditions without head motion ($\bar{\mathcal{M}}$) and affordance $F_{a}$ (source from 3D objects). The right part of Tab. 2 shows different encoding methods for the head motion, e.g. randomly initialize motion encoder (ri. $f_\mathcal{M}$), and use divided space-time attention (d. $f_{st}$) for extracting video features, also reflecting the impact of these modalities.
> | Metrics   | all modality | w/o $\bar{\mathcal{M}}$ | w/o $F_{a}$ | ri. $f_\mathcal{M}$ | d. $f_{st}$ |
> |:-----------:|:-----------:|:--------:|:-------:|:----------:|:-------:|
> | Precision   | 0.78        |   0.68   |   0.71  |   0.73     |   0.76  |
> | Recall      | 0.79        |   0.73   |   0.64  |   0.69     |   0.75  |
> | F1          | 0.76        |   0.69   |   0.66  |   0.70     |   0.75  |
> | geo. (cm)   | 12.62       |   19.86  |   19.13 |   14.57    |   13.04 |
> | AUC         | 78.02       |   74.36  |   75.21 |   76.05    |   77.88 |
> | aIOU        | 14.94       |   11.75  |   12.05 |   12.92    |   14.62 |
> | SIM         | 0.436       |   0.403  |   0.410 |   0.423    |   0.431 |

---

> > ### Comment · Reviewer_PyYF · 2024-08-12
> >
> > thanks for these clarifications, this answers my queries

---

### Author Rebuttal · Authors · 2024-08-05

Thanks to all the reviewers for their effort and constructive feedback. We are encouraged that the reviewers appreciate our work, including:

* The key idea of the method is ingenious for the proposed task [Reviewer paDU, fLMx]
* The superiority of performance over baselines [Reviewer PyYF, paDU, fLMx, L6qS]
* Extensive and comprehensive analysis of the proposed method [Reviewer PyYF, paDU, fLMx]
* Clear and well written [Reviewer PyYF, paDU]

There are questions prompting additional investigations. We provide here a pdf containing the results of those investigations. Its contents are:

* Figure 1: Statistical information of the dataset and schematic figure of modulation for reviewer fLMx
* Figure 2: Inference results of noisy 3D objects for reviewer L6qS
* Figure 3: The same interaction across videos for reviewer L6qS

Specific context and discussions are presented in the corresponding review rebuttals. We are open to addressing any issues from reviewers during the discussion stage. We will adapt the paper based on your insightful comments, feedback, and questions.

Many thanks,

The authors

---

### Decision · Program_Chairs · 2024-09-25

**Decision:**

Accept (poster)

**Comment:**

The paper presents a novel approach to estimating 3D human-object interaction regions from egocentric videos, which is a crucial yet under-explored area. The reviewers find the paper's presentation clear. The method is novel and experimental validations are comprehensive.
While some reviewers initially had concerns about the technical novelty and clarity of the contributions, these were adequately addressed during the rebuttal phase, leading to an increase in their ratings.

The proposed method's ability to integrate visual appearance, head motion, and 3D object information to infer interaction regions is seen as a significant contribution to the field.  Additionally, the creation of a new dataset with 3D contact and affordance annotations further strengthens the paper's impact.  Overall, the positive reviews, the authors' responsiveness to feedback, and the potential impact of the work justify accepting this paper.